# Positive Affect Moderates the Influence of Perceived Stress on the Mental Health of Healthcare Workers during the COVID-19 Pandemic

**DOI:** 10.3390/ijerph192013600

**Published:** 2022-10-20

**Authors:** Xu Wang, Rui Luo, Pengyue Guo, Menglin Shang, Jing Zheng, Yuqi Cai, Phoenix K. H. Mo, Joseph T. F. Lau, Dexing Zhang, Jinghua Li, Jing Gu

**Affiliations:** 1School of Public Health, Sun Yat-sen University, No.74, Zhongshan Second Road, Guangzhou 510080, China; 2Shenzhen Health Development Research and Data Management Center, Shenzhen 518028, China; 3Division of Behavioral Health and Health Promotion, The School of Public Health and Primary Care, Faculty of Medicine, The Chinese University of Hong Kong, Shatin, Hong Kong, China; 4The Chinese University of Hong Kong Shenzhen Research Institute, Shenzhen 518172, China; 5Centre for Medical Anthropology and Behavioral Health, Sun Yat-sen University, Guangzhou 510080, China; 6JC School of Public Health and Primary Care, The Chinese University of Hong Kong, Hong Kong, China; 7Sun Yat-sen University Global Health Institute, School of Public Health, Institute of State Governance, Sun Yat-sen University, Guangzhou 510080, China; 8Center for Health Information Research, Sun Yat-sen University, Guangzhou 510080, China

**Keywords:** positive affect, perceived stress, mental health, healthcare workers, COVID-19

## Abstract

The coronavirus disease 2019 (COVID-19) pandemic has posed a profound psychological impact on healthcare workers. However, the role of positive affect in moderating the effect of perceived stress on the psychological states of healthcare workers remains unknown. This study aimed to analyze the moderating effect of positive affect on the association between stress and the mental health of healthcare workers during the COVID-19 pandemic. This cross-sectional study evaluated the relationships between perceived stress (the Perceived Stress Scale), positive affect (the Positive and Negative Affect Schedule), depression (the Patient Health Questionnaire-9), and anxiety (the Generalized Anxiety Disorder 7-item Scale) during the COVID-19 pandemic in 644 Chinese healthcare workers who completed online self-reports. The results revealed a significant negative association between positive affect and psychological problems, including stress, depression, and anxiety. At the total group level, multiple regression analysis showed that positive affect alleviated the influence of perceived stress on depression, but no significant moderating effect was found for anxiety. In the subgroups divided by perceived stress, the moderating effect of positive affect on depression was only significant in healthcare workers with a high level of perceived stress. These results suggested that positive affect played a moderative role in alleviating the effect of stress on depression among healthcare workers, particularly those with a high level of stress, thus emphasizing the importance of positive affect as an intervention strategy for promoting the mental health of healthcare workers in the context of the ongoing COVID-19 pandemic.

## 1. Introduction

The coronavirus disease 2019 (COVID-19) poses a serious challenge to the global public health system. As of 27 June 2022, more than 500 million confirmed COVID-19 cases and 6 million related deaths had been reported to the World Health Organization [1]. Severe acute respiratory syndrome coronavirus 2, the causative agent of COVID-19, is mainly transmitted through the air and close contact, putting healthcare workers at higher risk of exposure [2]. Throughout the COVID-19 pandemic, both work intensity and the working hours of healthcare workers have increased [3], exerting enormous psychological pressure and inducing stress, anxiety, and depression [4,5]. A meta-analysis of 62 studies on the mental health of healthcare workers in 17 countries during the COVID-19 pandemic showed a combined prevalence of 26% for anxiety and 25% for depression [6]. A multi-centered online survey of 1563 Chinese healthcare workers during the COVID-19 pandemic also reported detectable rates of depression and anxiety of 50.7% and 44.7%, respectively [7]. In addition, even as COVID-19 case numbers have begun to decline, healthcare workers remain afflicted by long-term psychological problems [8]. Thus, investigations of the underlying psychological effects of the COVID-19 pandemic are needed to facilitate mental health interventions and prevention of psychological problems [9].

The uncertainty and hazards associated with public health emergencies represent substantial stressors that may lead to a series of psychological pressure reactions. Stress is a feeling experienced in situations where external demands exceed the resources available [10]. Stress can also be comprehended as an imbalance between an individual’s exposure to daily stressors and their coping ability [11,12]. Several studies have demonstrated that higher perceived stress is a risk factor for anxiety and depression [13,14,15]. During the COVID-19 pandemic, the levels of perceived stress among healthcare workers have significantly increased due to increased burnout, lack of medical resources, a high risk of infection, and occupational stress [16,17,18]. People are more likely to experience stress and develop severe psychological problems during long-term negative events [19].

Positive affect is defined as an individual’s pleasant interpretation of and reaction to their surroundings, including positive emotional states such as confidence and enthusiasm [20]. A former study has shown that the negative effects of perceived stress depend on the surrounding environment as well as internal and external resources [21]. Positive affect, as a vital psychological resource, may play a moderating role in the relationship between perceived stress and psychological problems. The broaden-and-build theory proposes that positive affect can broaden the scope of attention and cognition and prompt psychological resilience in the presence of stressors, thus improving resistance to psychological disorders [22,23,24]. Moreover, positive affect can buffer the effects of stress by reducing negative affect at the psychological level. Nelson et al. [25] suggested that positive and negative affect can occur at the same time under stressful events, where positive affect can mitigate negative affect responses to help individuals cope with stressors and reduce stress responses by re-evaluating stressors as challenges (rather than threats or harms).

To date, most studies have focused on the negative impacts of negative affect on physical and mental health status or the simple relationships between positive affect and anxiety and depression [25,26,27]. Stress can cause a number of psychological problems, such as depression and anxiety [28,29]. However, few studies have investigated how positive affect moderates the effects of stress on psychological problems. Sewart et al. [30] found that positive affect moderates the positive relationship between chronic interpersonal stress and anxiety and depression in adolescents: students with high positive affect have a lower association between chronic interpersonal stress and major depressive disorder and social anxiety disorder, as compared with those with a low level of positive affect. Therefore, positive affect may be a crucial psychological resource for healthcare workers to mitigate the effects of chronic stress related to the COVID-19 pandemic. To the best of our knowledge, the impact of positive affect on the association between stress and mental health of healthcare workers during the COVID-19 pandemic has not been investigated.

Given the relationship between positive affect and perceived stress, depression, and anxiety, the purpose of this research was to evaluate the characteristics of positive affect and its moderating effect on the relationship between perceived stress and anxiety, and depression. Moreover, in order to further explore the differences in the moderating role of positive affect under different stress statuses, we divided the participants into high and low levels of stress subgroups. Two hypotheses were defined, and we expected:

At the total group level, positive affect could alleviate the impacts of perceived stress on psychological problems, including depression and anxiety.At the subgroup level, the moderating role of positive affect was different according to the level of perceived stress. Particularly, compared with the subgroup with a low level of stress, the moderating effect of positive affect on the relationship between stress and psychological problems was more significant in the high-stress subgroup.

## 2. Methods

### 2.1. Study Procedure

Recruitment information for this cross-sectional study was delivered at all levels of medical institutions, including general hospitals, specialized hospitals, traditional Chinese medicine hospitals, Centers for Disease Control and Prevention, community health service centers, and maternal and child health care hospitals in all municipal districts of Guangzhou, Guangdong province, China. The study assistants sent the online questionnaire using the Questionnaire Star survey tool (www.wjx.cn, accessed on 6 December 2021) to collect all data.

### 2.2. Participants

Healthcare workers could participate in the study by scanning the QR code and providing their contact information, including mobile phone numbers and social media accounts such as QQ and WeChat. Participants were then informed of the background, significance, purpose, and confidentiality of this study and were required to complete and sign electronic informed consent forms.

The inclusion criteria for potential participants were as follows: (1) at least 18 years old; (2) currently employed as a healthcare worker (including clinicians, nurses, and public health personnel); (3) able to independently complete the online questionnaire; and (4) having a mobile phone or tablet connected to the Internet. Potential participants were excluded if they reported a serious mental disorder or suicidal ideation. This study was approved by the Public Health Ethics Committee of Sun Yat-sen University [2021-120].

### 2.3. Measurements

#### 2.3.1. Sociodemographic Information

Participants provided background characteristics, including age, gender, education level, income, marital status, residence status, seniority, and professional title.

#### 2.3.2. The Positive and Negative Affect Schedule

The Positive and Negative Affect Schedule was developed by Watson and Clark based on a two-dimensional model of emotions: positive affect (10 items, i.e., attentive, interested, alert, excited, determined, strong, active, enthusiastic, inspired, and proud) and negative affect (10 items, i.e., irritable, upset, distressed, hostile, scared, ashamed, guilty, nervous, jittery and afraid) [31]. Each item is rated on a 5-point scale from 1 = “not at all” to 5 = “strongly.” Our study used the positive affect subscale of the Positive and Negative Affect Schedule, which reflected the extent to which an individual felt positive emotions such as “enthusiastic” and “active.” The Cronbach’s alpha coefficient for the scale was 0.920 in the present study.

#### 2.3.3. Perceived Stress Scale

The perceived stress scale was used to measure the extent of self-perceived stress over the past month [32]. The scale consists of 10 items rated on a 5-point Likert scale from 0 = “never” to 4 = “very often.” Four positively stated items are reversely encoded and summed with the other items. Total scores range from 0 to 40, with higher scores indicating higher levels of perceived stress. The details of the scale are shown in Appendix A. Participants with cutoff scores of >=15 for the perceived stress scale as “high-stress group” and <15 as “low-stress group” [33]. The Cronbach’s alpha coefficient for the scale was 0.854 in the present study.

#### 2.3.4. Generalized Anxiety Disorder 7-Item Scale

The Generalized Anxiety Disorder 7-item Scale was used to assess the severity of anxiety symptoms reported by participants during the past 2 weeks. The scale has seven items, which are rated on a 4-point Likert scale from 0 = “not at all” to 3 = “nearly every day.” The total scores range from 0 to 21. The details of the items on the scale are shown in Appendix A. The Generalized Anxiety Disorder 7-item Scale has shown reliability and validity in people with anxiety [34], and Cronbach’s alpha coefficient for this scale was 0.934 in this study.

#### 2.3.5. Patient Health Questionnaire-9

The Patient Health Questionnaire-9 is a self-rating scale widely used in primary health care to evaluate depression in the past 2 weeks [35]. The Patient Health Questionnaire-9 includes nine items rated on a 4-point Likert scale from 0 = “not at all” to 3 = “nearly every day,” with total scores ranging from 0 to 27. The details of the scale are shown in Appendix A. The PHQ-9 scale has been shown to have good reliability and validity [36]. The Cronbach’s alpha coefficient for the scale was 0.897 in this study.

### 2.4. Statistical Analysis

Descriptive analysis was performed by means of reporting frequencies and percentages for categorical variables and means and standard deviations (SD) for continuous variables. An Analysis of Variance and Chi-square tests were used to assess differences in perceived stress, positive affect, anxiety, and depression according to the general sociodemographic characteristics. The Bonferroni adjustment was used to account for multiple comparisons. Pearson’s correlation coefficient was used to assess the relationships between variables. 

Multiple regression analysis was performed to determine whether positive affect moderates the influence of stress on the two psychological problems (depression and anxiety). To increase the interpretability of the moderation model and control for multicollinearity, stress and positive affect were centered [37], and then the product term was calculated. The variables were entered in the following steps: Model1) the controlled sociodemographic variables, Model2) perceived stress, Model3) positive affect, and Model4) the interaction term of stress × positive affect. The existence of a moderating effect was judged by the statistical significance of the interaction term and the changes in *R*^2^ and *F* in the final model. Significant positive affect × perceived stress interactions were also explored using the simple slope post hoc analyses. According to Jaccard et al. [38], the simple slope analysis was performed to verify the interaction effect by determining the slopes of the regression lines at low (1 SD below mean), intermediate (mean), and high (1 SD above mean) values of positive affect differed significantly from zero. In addition, similar analyses were performed for the two subgroups with different stress levels parallelly. All analyses were conducted using R version 4.0.3 (R Foundation for Statistical Computing).

## 3. Results

### 3.1. Descriptive Analyses

The descriptive statistics and differences in stress, positive affect, anxiety, and depression according to sociodemographic characteristics are shown in Table 1. 

The average age and seniority of the 644 participants were 30.93 ± 6.95 and 8.11 ± 7.77 years. As shown in Table 1, there were more men (71.4%) than women. About half (47.4%) of the participants had a monthly income ranging from 5000 to 10,000 yuan, were married (48.6%), and had a junior title (49.2%). Most of the healthcare workers possessed a college degree or above (85.1%) and were not living alone at the time of the survey (78.5%). In addition, Positive affect showed statistically significant differences among participants grouped by monthly income (*F* = 4.717, *p* = 0.009) and marital status (*F* = 4.968, *p* = 0.007). Specifically, the level of positive affect was higher in participants with incomes >10,000 yuan and those who were married. Depression levels significantly differed among participants grouped by marital status (*F* = 3.916, *p* = 0.020) and residential status (*t* = 2.660, *p* = 0.008); the level of depression was higher in participants who were unmarried or living alone. The difference in anxiety levels among participants grouped by education level was significant, but the pairwise comparisons did not reach significance. No statistically significant differences in stress were detected among participants grouped by sociodemographic characteristics.

### 3.2. Average Scores and Correlations among Mental Health Measurements

The means, SDs, and correlation coefficient for the studied mental health status are presented in Table 2. The average scores on the perceived stress, positive affect, anxiety, and depression scales were 17.08 ± 6.60, 27.16 ± 6.16, 7.46 ± 4.70, and 9.26 ± 5.01, respectively. Perceived stress was negatively correlated with positive affect (*r* = −0.47, *p* < 0.001) and positively correlated with anxiety and depression (*r* = 0.73, *p* < 0.001 and *r* = 0.71, *p* < 0.001, respectively). Positive affect was negatively correlated with anxiety and depression (*r* = −0.39, *p* < 0.001 and *r* = −0.40, *p* < 0.001, respectively). 

### 3.3. Multiple Regression Analyses in the Total Group

Table 3 shows the results of the moderated role of positive affect on the relationship between stress and depression in the total group. After controlling for the sociodemographic variables, positive affect was negatively associated with depression (*b* = −0.075, *p* = 0.004, Model 3), and perceived stress was positively associated with depression (*b* = 0.503, *p* < 0.001, Model 3). The interaction term stress × positive affect was significantly negatively associated with depression in Model4 (*b* = −0.010, *p* = 0.002). The simple slope analysis revealed that compared with participants with a low level of positive affect (1 SD below mean, *B* = 0.571, *p* < 0.001), participants with a high level of positive affect (1 SD above mean, *B* = 0.448, *p* < 0.001) had a weaker association between perceived stress and depression (Figure 1a).

Similar analyses were conducted for anxiety. As shown in Table 4, positive affect was negatively associated with anxiety (*b* = −0.055, *p* = 0.019, Model 3), and perceived stress was positively associated with anxiety (*b* = 0.501, *p* < 0.001, Model3). However, the interaction term stress × positive affect showed there was no moderating effect of positive affect between perceived stress and anxiety in the total group (*b* = −0.003, *p* = 0.307, Model4). In conclusion, this suggested that positive affect had a moderating effect on the relationship between stress and depression rather than anxiety in the total group.

### 3.4. Multiple Regression Analyses in the Subgroups

Table 5 shows the results of moderated multiple regression analysis on the relationship between stress and depression in the high-stress subgroup (*n* = 433, average score of perceived stress: 20.96 ± 4.26). After controlling for the sociodemographic variables, positive affect was negatively associated with depression (*b* = −0.111, *p* < 0.001, Model3), and perceived stress was positively associated with depression (*b* = 0.535, *p* < 0.001, Model3). The interaction term stress × positive affect was significantly negatively associated with depression in Model4 (*b* = −0.017, *p* = 0.024), suggesting that positive affect had a moderating effect on depression in the high-stress group. As shown in Figure 1b, similar to the results of the total group, the moderating effect of positive affect was observed for the high-stress group (1 SD below mean, *B* = 0.610, *p* < 0.001, 1 SD above mean, *b* = 0.425, *p* < 0.001). This suggested that positive affect could moderate the impact of stress on depression in the high-stress subgroup.

Appendix A showed the results of moderating effect of positive affect on the relationship between stress and depression in the low-stress subgroup (*n* = 231, average score of perceived stress: 9.82 ± 3.21). Positive affect had no significant association with depression (*b* = −0.026, *p* = 0.516, Model3). There was also no moderating effect of positive affect was observed (*b* = −0.002, *p* = 0.906, Model4).

The results of moderating effect of positive affect on the relationship between stress and anxiety in the high- and low-stress subgroups were presented in Appendix A, respectively. In the high-stress group, the interaction term stress × positive affect showed there was no moderating effect of positive affect between perceived stress and anxiety (*b* = −0.004, *p* = 0.525, Model4 in Appendix A). In the low-stress group, positive affect had no significant association with anxiety (*b* = −0.008, *p* = 0.829, Model3 in Appendix A). There was also no moderating effect of positive affect observed (*b* = 0.006, *p* = 0.591, Model4 in Appendix A).

## 4. Discussion

To the best of our knowledge, this is the first study to analyze the relationship between positive affect and perceived stress and psychological problems among healthcare workers in the context of the COVID-19 pandemic. We found that positive affect was an important moderator, alleviating the impact of perceived stress on depression, especially among healthcare workers with a high level of stress.

According to the present study, healthcare workers in marital relationships had higher levels of positive affect and lower levels of depression than unmarried one. According to the social support theories, the marriage relationship is the most potent source of social support and a significant form of social relationship, which could have a positive impact on health and emotional well-being [39,40]. Numerous practical studies consistently showed that relationships such as marriage could help to reduce the risks of mental disorders, including depression specifically [41,42,43]. Moreover, there was a higher positive affect among healthcare workers with better financial states in this study. It is consistent with previous research, which found that individuals with low socioeconomic backgrounds were more likely to report lower positive affect compared with the high socioeconomic [44,45]. This may be due to increased psychological pressure resulting from potential socioeconomic challenges faced during the pandemic, which may seriously impact mental health, including positive affect [46]. However, our findings revealed that the anxiety level of healthcare workers increased with a higher level of education. This result could be because better knowledge and understanding of the disease could engender stress and anxiety [47,48]. Therefore, these results showed that the external environment was closely related to mental health and psychological regulation function, which reminds health managers that it is effective to enhance positive affect and mental health by considering marital status, level of education, and financial state of healthcare workers together.

This study found that positive affect was a protective factor for both anxiety and depression in healthcare workers during the COVID-19 pandemic. Such finding was consistent with literature which reported that positive affect was negatively associated with mental health problems and recovery time from stressful events [49,50]. Furthermore, the present results demonstrated that positive affect mitigated the relationship between perceived stress and depression, and such a moderating effect was significant in participants with a high level of perceived stress. Thong et al. found that positive affect can buffer the association between pain intensity and depressive symptoms [51]. A longitudinal study used hierarchical multilevel modeling to validate positive affect as a protective factor for reducing the effects of chronic stress on depression among high school students in the United States [30]. Moreover, according to the broaden-and-build theory, during periods of stress, positive affect enables individuals to develop abilities, such as knowledge or social relations, that could play a protective role during high-stress situations [27]. Such a protective effect was stronger in healthcare workers with a high level of stress than those with a low level of stress, reminding us that psychological problems could be reduced and improved via positive affect intervention on the premise of identification of the high level of stress of healthcare workers [52,53].

However, positive affect did not have a moderating effect on perceived stress and anxiety in the present study. Such a finding suggests that anxiety is, to some extent, more difficult to alleviate through positive affect than depression. It is possible that according to the influential tripartite [54] and quadripartite model [55], a lack of positive affect was the core component for depression [56], whereas anxiety was related to personal trait [57,58] and was more likely to be influenced by external factors such as heavy workload of healthcare duties during COVID-19 [59,60]. Therefore, given that the relationship between stress and anxiety could not be reduced by positive affect, interventions such as emotional freedom techniques [61] and mindfulness-based breathing therapy practices [62] can be formulated to mitigate the level of anxiety among healthcare workers. 

Because of the nature of healthcare work, healthcare workers were more likely to encounter epidemic-related stressors and develop psychological problems such as depression after the outbreak of COVID-19 than the general population [63,64]. This situation has been reported for almost all healthcare professional activities, such as dentistry [65], general surgery [66], ophthalmology [67], and nursing [61]. 

The present results indicate that positive affect could alleviate the influence of perceived stress on depression in healthcare workers. This finding suggests that positive affect plays a crucial role in promoting mental health and preventing psychological problems related to COVID-19-induced stress in healthcare workers. Similarly, a meta-analysis of 51 studies highlighted the importance of positive emotional intervention strategies for mental health. Interventions that foster positive affect was shown to significantly enhance well-being and reduce depression severity [53]. Furthermore, Coifman et al. [68] demonstrated that 3–6-min mental interventions, such as expressive writing and adaptive emotion regulation activities, can boost positive affect in medical and emergency-response personnel. However, several studies indicated that healthcare workers experienced low levels of positive affect and high levels of negative affect, such as fear or helplessness during the COVID-19 [18,69]. They were also reported that have low participation rates in psychological intervention programs [70] and tended to use emotional suppression behaviors when facing COVID-19 patients, which may increase their level of psychological stress and trigger psychological problems [26]. In our study, positive affect was a significant factor in alleviating the association between stress and depression, particularly in participants with high stress levels. In conclusion, this study provided important insight for the future development of simple and effective interventions based on the stress levels of healthcare workers to promote positive affect and attenuate psychological problems in the context of the COVID-19 pandemic. 

This study has several limitations. First, because the study design involved online non-probability sampling and the participants were from a single region (Guangzhou), the representativeness and extrapolation of the results may be limited to some extent. Second, the average age of healthcare workers in our research was relatively low and may not correctly represent the population of healthcare workers. Thirdly, our study was cross-sectional and thus could not determine the causal relationship between variables. Consequently, longitudinal research is warranted to determine the dynamics of the relationships between positive affect, stress, depression, and anxiety.

## 5. Conclusions

The mental health of healthcare workers in the context of COVID-19 deserves immediate attention. The findings of the present study showed that positive affect played a significant role in alleviating the effect of stress on depression among healthcare workers, particularly those with high levels of stress. As a result, our findings suggest that positive affect may be a significant perspective for health managers to design targeted interventions such as expressive writing and adaptive emotion regulation activities to mitigate the effects of perceived stress on psychological problems in healthcare workers during COVID-19.

## Figures and Tables

**Figure 1 ijerph-19-13600-f001:**
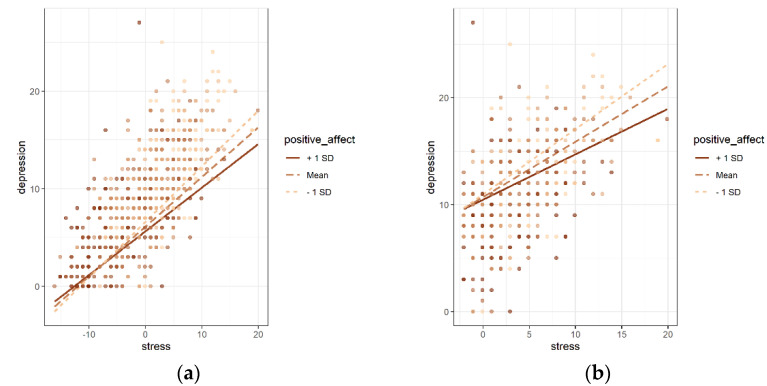
The moderating effect of positive affect in the total group (**a**) and high−stress group (**b**).

**Table 1 ijerph-19-13600-t001:** Descriptive relationships of sociodemographic factors with stress, positive affect, depression, and anxiety (*N* = 644).

Characteristics	Total	Positive Affect		Stress		Depression		Anxiety	
Variables	*N* (%)/Mean ± SD	Mean ± SD	*p*	Mean ± SD	*p*	Mean ± SD	*p*	Mean ± SD	*p*
Gender			0.215		0.654		0.759		0.815
Male	190 (28.6)	27.6 (6.37)		17.3 (6.76)		9.36 (5.02)		7.53 (4.45)	
Female	474 (71.4)	27.0 (6.07)		17.0 (6.54)		9.23 (5.02)		7.43 (4.81)	
Education level			0.513		0.183		0.065		0.036 *
Junior high or below	2 (0.3)	30.0 (11.3)		8.50 (7.78)		1.00 (0.00)		0.00 (0.00)	
High school and vocational school	97 (14.6)	26.6 (6.49)		17.1 (6.70)		9.24 (5.39)		6.93 (4.62)	
College or above	565 (85.1)	27.2 (6.09)		17.1 (6.57)		9.30 (4.94)		7.58 (4.70)	
Individual monthly income (yuan)			0.009 **, a < c		0.089		0.113		0.930
<5000 ^a^	146 (22.0)	26.0 (6.04)		17.8 (6.52)		10.0 (5.29)		7.58 (4.92)	
5000–10,000 ^b^	315 (47.4)	27.1 (5.85)		17.3 (6.57)		9.14 (4.89)		7.40 (4.46)	
>10,000 ^c^	203 (30.6)	28.0 (6.58)		16.3 (6.66)		8.92 (4.98)		7.46 (4.93)	
Marital status			0.007 **, a < b		0.163		0.020 *, a > b		0.796
Single ^a^	330 (49.8)	26.4 (5.82)		17.6 (6.48)		9.82 (4.89)		7.60 (4.37)	
Currently married ^b^	322 (48.6)	27.9 (6.34)		16.7 (6.64)		8.79 (5.07)		7.36 (4.98)	
Divorced or other ^c^	10 (1.5)	28.6 (4.84)		15.5 (7.52)		7.80 (5.03)		7.70 (5.77)	
Status of residence			0.115		0.433		0.008 **		0.095
Living alone	143 (21.5)	26.4 (6.35)		17.5 (6.28)		10.3 (5.12)		8.03 (4.60)	
Not living alone	521 (78.5)	27.4 (6.10)		17.0 (6.69)		8.99 (4.95)		7.30 (4.72)	
Professional title			0.730		0.305		0.757		0.408
Junior title	327 (49.2)	27.0 (5.99)		16.7 (6.20)		9.27 (4.88)		7.21 (4.46)	
Intermediate title	195 (29.4)	27.4 (6.27)		17.7 (7.14)		9.09 (5.26)		7.67 (5.12)	
Senior title	142 (21.4)	27.2 (6.40)		17.1 (6.73)		9.50 (5.01)		7.75 (4.64)	

*: *p* < 0.05; **: *p* < 0.01; In the Individual monthly income variable, “^a^” is the <5000 yuan category, “^b^” is the 5000-10,000 yuan category, “^c^” is the >10,000 yuan category; In the Marital status variable, “^a^” is the Single category, “^b^” is the Currently married category, “^c^” is the Divorced or other category.

**Table 2 ijerph-19-13600-t002:** Description and Correlations of key variables (*N* = 644).

Variables	Variable Description Mean ± SD	Stress	Positive Affect	Anxiety	Depression
Stress	17.08 ± 6.60	1.00			
Positive affect	27.16 ± 6.16	−0.47 ***	1.00		
Anxiety	7.46 ± 4.70	0.73 ***	−0.39 ***	1.00	
Depression	9.26 ± 5.01	0.71 ***	−0.40 ***	0.84 ***	1.00

***: *p* < 0.001.

**Table 3 ijerph-19-13600-t003:** Moderating effect of positive affect on the relationship between stress and depression in the total group.

	Model1		Model2		Model3		Model4	
	*b*	*p*	*b*	*p*	*b*	*p*	*b*	*p*
Gender (Female)	−0.046	0.915	0.045	0.884	−0.009	0.976	−0.040	0.894
Age	−0.014	0.794	−0.006	0.885	−0.015	0.695	−0.027	0.483
Education level (Junior high or below)								
High school and vocational school	8.652	0.017 *	3.434	0.179	3.400	0.181	3.746	0.138
College or above	9.092	0.011 *	3.683	0.147	3.671	0.146	4.009	0.110
Marital status (Single)								
Married	−0.456	0.400	0.046	0.904	0.141	0.712	0.177	0.642
Divorced or other	−1.610	0.348	−0.406	0.738	−0.284	0.814	−0.236	0.844
Status of residence (Not living alone)	−1.006	0.055	−0.921	0.013 *	−0.889	0.016 *	−0.942	0.010 *
Income (yuan, <5000)								
5000–10,000	−0.883	0.098	−0.471	0.211	−0.406	0.280	−0.440	0.239
>10,000	−0.893	0.152	0.002	0.997	0.082	0.852	0.026	0.952
Professional title (Junior title)								
Intermediate title	0.557	0.274	−0.429	0.237	−0.392	0.277	−0.464	0.196
Senior title	0.578	0.274	0.143	0.703	0.186	0.617	0.132	0.720
Seniority	−0.013	0.771	−0.016	0.610	−0.013	0.667	−0.002	0.955
Stress (A)			0.536	<0.001 ***	0.503	<0.001 ***	0.509	<0.001 ***
Positive affect (B)					−0.075	0.004 **	−0.078	0.002 **
A × B							−0.010	0.002 **
Model *F* (*p*)	1.862 (0.036)		53.42 (<0.001)		50.79 (<0.001)		48.71 (<0.001)	
*R*^2^ (Δ *R*^2^)	0.033		0.517 (0.484)		0.524 (0.007)		0.531 (0.007)	
Δ *F* (*p*)			649.79 (<0.001)		8.540 (0.004)		9.867 (0.002)	

*: *p* < 0.05; **: *p* < 0.01; ***: *p* < 0.001. The variance inflation factor (VIF) of Model4 was 1.01–1.77 (<10). Model1: the regression with sociodemographic variables. Model2: the regression with perceived stress after controlling the sociodemographic variables. Model3: the regression with perceived stress and positive affect after controlling the sociodemographic variables. Model4: the regression with interaction term of stress × positive affect after controlling the sociodemographic variables.

**Table 4 ijerph-19-13600-t004:** Moderating effect of positive affect on the relationship between stress and anxiety in the total group.

	Model1		Model2		Model3		Model4	
	*b*	*p*	*b*	*p*	*b*	*p*	*b*	*p*
Gender (Female)	0.026	0.948	0.115	0.678	0.075	0.784	0.066	0.810
Age	−0.022	0.664	−0.014	0.691	−0.021	0.551	−0.024	0.488
Education level (Junior high or below)								
High school and vocational school	7.204	0.034 *	2.095	0.366	2.069	0.370	2.173	0.347
College or above	7.948	0.019 *	2.651	0.249	2.642	0.249	2.743	0.232
Marital status (Single)								
Married	0.077	0.880	0.569	0.102	0.638	0.067	0.649	0.063
Divorced or other	0.186	0.909	1.365	0.215	1.454	0.186	1.468	0.181
Status of residence (Not living alone)	−0.808	0.102	−0.724	0.031 *	−0.701	0.037 *	−0.717	0.033 *
Income (yuan, <5000)								
5000–10,000	−0.333	0.508	0.070	0.839	0.118	0.730	0.108	0.753
>10,000	−0.328	0.576	0.548	0.172	0.607	0.130	0.590	0.141
Professional title (Junior title)								
Intermediate title	0.753	0.117	−0.211	0.520	−0.184	0.573	−0.206	0.530
Senior title	0.806	0.106	0.379	0.263	0.411	0.224	0.395	0.243
Seniority	−0.008	0.851	−0.011	0.702	−0.009	0.751	−0.005	0.847
Stress (A)			0.525	<0.001 ***	0.501	<0.001 ***	0.503	<0.001 ***
Positive affect (B)					−0.055	0.019 *	−0.056	0.017 *
A × B							−0.003	0.307
Model *F* (*p*)	1.20 (0.276)		60.68 (<0.001)		57.14 (<0.001)		53.40 (<0.001)	
*R*^2^ (Δ *R*^2^)	0.022		0.549 (0.527)		0.553 (0.004)		0.554 (0.001)	
Δ *F* (*p*)			757.58 (<0.001)		5.533 (0.019)		1.045 (0.307)	

*: *p* < 0.05; ***: *p* < 0.001. The variance inflation factor (VIF) of Model4 was 1.01–1.93 (<10). Model1: the regression with sociodemographic variables. Model2: the regression with perceived stress after controlling the sociodemographic variables. Model3: the regression with perceived stress and positive affect after controlling the sociodemographic variables. Model4: the regression with an interaction term of stress × positive affect after controlling the sociodemographic variables.

**Table 5 ijerph-19-13600-t005:** Moderating effect of positive affect on the relationship between stress and depression in the high-stress group.

	Model1		Model2		Model3		Model4	
	*b*	*p*	*b*	*p*	*b*	*p*	*b*	*p*
Gender (Female)	0.209	0.661	0.219	0.582	0.079	0.843	0.081	0.837
Age	0.052	0.345	0.017	0.709	−0.004	0.929	−0.010	0.832
Education level (High school and vocational school)								
College or above	0.077	0.901	−0.055	0.916	−0.131	0.799	0.037	0.943
Marital status (Single)								
Married	−0.399	0.510	−0.237	0.641	−0.138	0.784	0.002	0.997
Divorced or other	−1.609	0.453	−2.008	0.265	−1.959	0.271	−1.801	0.309
Status of residence (Not living alone)	−1.077	0.056	−1.338	0.005 **	−1.307	0.005 **	−1.369	0.003 **
Income (yuan, <5000)								
5000–10,000	−1.067	0.063	−0.708	0.140	−0.614	0.197	−0.713	0.134
>10,000	−0.211	0.761	0.038	0.949	0.119	0.836	0.020	0.972
Professional title (Junior)								
Intermediate title	0.644	0.243	−0.202	0.666	−0.155	0.736	−0.223	0.629
Senior title	0.096	0.868	−0.315	0.518	−0.307	0.524	−0.325	0.498
Seniority	−0.028	0.532	0.020	0.591	0.031	0.409	0.035	0.355
Stress (A)			0.568	<0.001 ***	0.535	<0.001 ***	0.487	<0.001 ***
Positive affect (B)					−0.111	<0.001 ***	−0.044	0.317
A × B							−0.017	0.024 *
Model *F* (*p*)	1.10 (0.359)		16.32 (<0.001)		16.27 (<0.001)		15.62 (<0.001)	
*R*^2^ (Δ *R*^2^)	0.028		0.318 (0.290)		0.336 (0.018)		0.344 (0.008)	
Δ *F* (*p*)			178.68 (<0.001)		10.992 (<0.001)		5.106 (0.024)	

*: *p* < 0.05; **: *p* < 0.01; ***: *p* < 0.001. The variance inflation factor (VIF) of Model4 was 1.03–1.81 (<10). Model1: the regression with sociodemographic variables. Model2: the regression with perceived stress after controlling the sociodemographic variables. Model3: the regression with perceived stress and positive affect after controlling the sociodemographic variables. Model4: the regression with an interaction term of stress × positive affect after controlling the sociodemographic variables.

## Data Availability

The raw data supporting the conclusions of this article will be made available by the authors, without undue reservation.

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
