# Peer review of "Positive Affect Moderates the Influence of Perceived Stress on the Mental Health of Healthcare Workers during the COVID-19 Pandemic"

_ijerph, 2022, doi:10.3390/ijerph192013600_

Round 1

Reviewer 1 Report

REVIEW: The positive affect moderates the influence of perceived stress on the mental health of healthcare workers during the COVID-19 pandemic

I was happy to review this interesting piece of research, due to the emphasis on the positive affect in copying with the stress caused by taking care of patients in the present era of COVID-19 pandemic. I would like to see this paper as published after some minor revisions.

Abstract

The variable used in making subgroups should be mentioned already on the line 37. For the time being, the reader must go to as long as to Introduction, lines 103-104, to get a proper explanation of the subgroups mentioned.

1. Introduction

I see that this section gives a thorough picture of the background, the rationale, and the significance of this particular research. A detail: Line 93 ends up with a notion of ‘positive effect’. Should it be ‘positive affect’?

2. Methods

In section 2.3.1. I wonder what the meaning of the expression ‘Technical title’ is. Judging from the following Table 1, it could mean a hierarchical position, like an employee, a supervisor, an executive etc.  or the length of time served in health care, from an apprentice to a senior worker. Please, explain, or choose another word to catch the organizational and cultural meaning of ‘Technical title’.

I would prefer to have all the items of all the scales to be listed in sections 2.3.2. – 2.3.5. This would give the reader a comprehensive picture of the psychological phenomena used to measure the research objects. For now, the reader must check the items in the articles mentioned in the references.

3. Results

I see that the results of a quantitative research should not be understood as tables filled with computed numerical figures, but rather, the results are what the researchers make of their figures.

I suggest a traditional way to formulate this section: Firstly, a short introduction is written about the research question to be presented in Table X /Figure Y, secondly Table X/Figure Y is inserted, and thirdly, explanatory and interpretitive comments follow, emphasizing the main points gathered from the numerical figures presented. This would mean a technical revision of Results -section. It is not a question of the contents, but about how to present it for readers to catch the starting point of the message of the research to be discussed further in the next section.

4. Discussion

I find that this section is properly themed around these new findings, earlier research, and the limitations of this research.

5. Conclusions

I suggest that the authors raise their level of ambition and give some examples of potential interventions to alleviate the effects of perceived stress with positive affect applications. Also, conclusions and practical implications should be formulated for those health care workers, whose anxiety could not be relieved by their own positive affect.

All the best for your revision work, your Reviewer.

Reviewer 2 Report

The topic is important and the structure of the paper is designed well.

I couldn't find which part answer this analysis "According to Jaccard et al. [36],the interaction effect was verified via determining the slopes of the regression lines at low (1 SD below mean), intermediate (mean) and high (1 SD above mean) values of positive affect differed significantly from zero. (lines 177-180)"Please make it clearly this point.

Good luck.

Reviewer 3 Report

Dear Authors,

generally, the write-up of the paper is good. Here follow some suggestion in order to improve the manuscript.

Research hypotheses shall be declared at the end of the Introduction, they should be labeled as such, and provide a numerical listing of each hypothesis (even just one). This listing is key to the paper. The same sequence of hypothesis testing will be used to structure the Materials and Methods, Results, Discussion, and the Conclusions sections.

Introduction length is a bit too much but is ok.

Discussion is relatively short and should be implemented (see comments below) in contents (rather than in length).

Lines 313-5

The authors could highlight that epidemic-related stressors may be encountered in every field of healthcare.

Examples should be outlined. The authors could add the following sentence to the paragraph:

“Because of the nature of healthcare work, healthcare workers were more likely to encounter epidemic-related stressors and develop psychological problems like depression after the outbreak of COVID-19 than general population [49,50]. This situation has been reported for quite all the healthcare professional activity such as dentistry [Paolone G, Mazzitelli C, Formiga S, Kaitsas F, Breschi L, Mazzoni A, Tete G, Polizzi E, Gherlone E, Cantatore G. One-year impact of COVID-19 pandemic on Italian dental professionals: a cross-sectional survey. Minerva Dent Oral Sci. 2022 Aug;71(4):212-222. doi: 10.23736/S2724-6329.21.04632-5. Epub 2021 Dec 1. PMID: 34851068.], general surgery [Mavrogenis AF, Scarlat MM. Stress, anxiety, and burnout of orthopaedic surgeons in COVID-19 pandemic. Int Orthop. 2022 May;46(5):931-935. doi: 10.1007/s00264-022-05393-2. PMID: 35384468; PMCID: PMC8984066.], ophthalmology [Grover R, Dua P, Juneja S, Chauhan L, Agarwal P, Khurana A. "Depression, Anxiety and Stress" in a Cohort of Registered Practicing Ophthalmic Surgeons, Post Lockdown during COVID-19 Pandemic in India. Ophthalmic Epidemiol. 2021 Aug;28(4):322-329. doi: 10.1080/09286586.2020.1846757. Epub 2020 Nov 13. PMID: 33185487.], nursery [Dincer B, Inangil D. The effect of Emotional Freedom Techniques on nurses' stress, anxiety, and burnout levels during the COVID-19 pandemic: A randomized controlled trial. Explore (NY). 2021 Mar-Apr;17(2):109-114. doi: 10.1016/j.explore.2020.11.012. Epub 2020 Dec 3. PMID: 33293201; PMCID: PMC7834511.], etc.

The authors did a very interesting job and collected interesting data.

This data, even if not significant should be discussed in the Discussion.

In particular:

Discussion.

Please add in the discussion a paragraph about the eventual influences of marital status.

Please discuss eventual articles in agreement or not on this topic.

Discussion.

Please add in the discussion a paragraph about the eventual influences of the annual income  and educational level.

Please discuss eventual articles in agreement or not on this topic.

Limitations.

Within the limitation of this study, please also mention that the average age is quite low and may not represent correctly all the healthcare professionals.

Discussion.

At the end of the discussion, please provide insight as to what types of research need to be done as a consequence of the knowledge found in your research.

Conclusions.

The authors could consider using a bullet point formatted conclusion, although not mandatory.

Round 2

Reviewer 3 Report

The authors have provided all the suggested improvements to the revised manuscript